# Incidence and Impact of Myocarditis in Genetic Cardiomyopathies: Inflammation as a Potential Therapeutic Target

**DOI:** 10.3390/genes16010051

**Published:** 2025-01-04

**Authors:** Yulia Lutokhina, Elena Zaklyazminskaya, Evgeniya Kogan, Andrei Nartov, Valeriia Nartova, Olga Blagova

**Affiliations:** 1Institute of Clinical Medicine, V.N. Vinogradov Faculty Therapeutic Clinic, I.M. Sechenov First Moscow State Medical University (Sechenov University), 119991 Moscow, Russia; nartov31229@gmail.com (A.N.); lera.avershina@yandex.ru (V.N.); blagovao@mail.ru (O.B.); 2Laboratory of Medical Genetics, B.V. Petrovsky Russian Research Center of Surgery, 119991 Moscow, Russia; 3N.P. Bochkov Research Centre for Medical Genetics, 119991 Moscow, Russia; 4Institute of Clinical Morphology and Digital Pathology, I.M. Sechenov First Moscow State Medical University (Sechenov University), 119991 Moscow, Russia; koganevg@gmail.com

**Keywords:** primary cardiomyopathy, myocarditis, inflammation, hypertrophic cardiomyopathy, arrhythmogenic cardiomyopathy, dilated cardiomyopathy, restrictive cardiomyopathy, left ventricular non-compaction, heart failure, arrhythmias

## Abstract

Background: Myocardial disease is an important component of the wide field of cardiovascular disease. However, the phenomenon of multiple myocardial diseases in a single patient remains understudied. Aim: To investigate the prevalence and impact of myocarditis in patients with genetic cardiomyopathies and to evaluate the outcomes of myocarditis treatment in the context of cardiomyopathies. Methods: A total of 342 patients with primary cardiomyopathies were enrolled. The study cohort included 125 patients with left ventricular non-compaction (LVNC), 100 with primary myocardial hypertrophy syndrome, 70 with arrhythmogenic right ventricular cardiomyopathy (ARVC), 60 with dilated cardiomyopathy (DCM), and 30 with restrictive cardiomyopathy (RCM). The diagnosis of myocarditis was based on data from myocardial morphological examination or a non-invasive diagnostic algorithm consisting of an analysis of clinical presentation, anti-cardiac antibody (Ab) titres, and cardiac MRI. Results: The prevalence of myocarditis was 74.3% in ARVC, 56.7% in DCM, 54.4% in LVNC, 37.5% in RCM, and 30.9% in HCM. Myocarditis had a primary viral or secondary autoimmune nature and manifested with the onset or worsening of chronic heart failure (CHF) and arrhythmias. Treatment of myocarditis in cardiomyopathies has been shown to stabilise or improve patient condition and reduce the risk of adverse outcomes. Conclusions: In cardiomyopathies, the genetic basis and inflammation are components of a single continuum, which forms a complex phenotype. In genetic cardiomyopathies, myocarditis should be actively diagnosed and treated as it is an important therapeutic target.

## 1. Introduction

Myocardial disease represents one of the most promising and dynamic areas of modern cardiology. Of even greater interest is the combination of different myocardial diseases in a single patient, with particular emphasis on the combination of cardiomyopathies with myocarditis. Over the past 15 years, the frequency of the search term “cardiomyopathy and myocarditis” in international bibliographic databases has increased exponentially.

The increased susceptibility of genetically altered myocardium to myocarditis is a topic of active discussion in the scientific community. Furthermore, there is evidence that myocarditis may act as a trigger for the initiation of an abnormal genetic programme [1,2,3,4]. It was demonstrated that children with acute myocarditis have significantly more frequent mutations in genes associated with various cardiomyopathies, such as *BAG3*, *DSP*, *PKP2*, *RYR2*, *SCN5A*, and *TNNI3* [5]. There is evidence that patients with cardiomyopathies caused by pathogenic or likely pathogenic variants in the *DSP* gene, as well as *LAMA4* and *MyBPC3*, have signs of active myocarditis, including elevated cardiac-specific enzymes, typical inflammatory changes on magnetic resonance imaging (MRI) and positron emission tomography (PET-CT), and on morphological examination [6,7]. In the European Cardiomyopathy and Myocarditis—Long Term (CMY-LT) registry, a combination of cardiomyopathy and myocarditis was identified in 128 individuals (3.2%), although this subgroup was not subjected to separate analysis.

It has been demonstrated that, in arrhythmogenic right ventricular cardiomyopathy (ARVC), histological examination reveals active myocarditis in more than 40% of patients [8,9]. A number of researchers have hypothesised that myocarditis represents a form of “hot phase” in the development of this cardiomyopathy [10,11,12]. Furthermore, the death of cardiomyocytes as a consequence of apoptosis, accompanied by the presentation of their antigens (Ag) during the natural course of ARVC, may give rise to the development of secondary inflammation in the myocardium [13].

Data on myocarditis in patients with hypertrophic cardiomyopathy (HCM) are limited. A study was conducted by a group of Italian scientists led by A. Frustaci in 2007, which yielded the result that the frequency of myocarditis in HCM was 23.5%. It was observed that, among these patients, there were no clinically stable patients, which serves to emphasise the contribution of myocarditis to the worsening of the clinical course of HCM.

Dilated cardiomyopathy (DCM) can be a consequence of myocarditis, but primary, genetic forms of DCM are caused by mutations in a wide range of genes. In such cases, the role of superimposed myocarditis in the course of the disease may be significant. It has been demonstrated that in DCM resulting from pathogenic or likely pathogenic variants in the *DMD* and *DYSF* genes, the myocardium is susceptible to infection by Coxsackie virus due to disruption of the cardiomyocyte membrane structure, which facilitates rapid viral spread within the myocardium [14,15]. It has been documented that cases of viral myocarditis in patients with Duchenne myopathy and DCM have resulted in a rapid progression of systolic dysfunction and patient mortality [16].

Left ventricular non-compaction (LVNC) was included in the classification of cardiomyopathies only in 2008 [17]. This condition is also a favourable background for myocarditis of viral etiology (cardiotropic viruses [18,19], paramyxovirus [20], SARS-CoV-2 [21]). In all described clinical cases, the accession of myocarditis led to decompensation of chronic heart failure (CHF). There is also evidence that acute myocarditis in a patient with LVNC provoked the development of continuously recurrent ventricular tachycardia (VT) [22].

Restrictive cardiomyopathy (RCM) is the least prevalent of the various forms of cardiomyopathy. Nevertheless, there are also reports of RCM occurring in conjunction with myocarditis. For example, endomyocardial biopsy (EMB) results from 286 patients with the RCM phenotype demonstrated the presence of myocarditis in 19 cases, representing 6.6% of the total [23]. Another study demonstrated that 7% of patients with morphologically confirmed myocarditis exhibited evidence of RCM on echocardiography (EchoCG) [24]. Amyloidosis has been identified as the most prevalent underlying cause of secondary RCM. Concurrently, amyloid can exert a direct toxic effect on cardiomyocytes and function as a substrate for the development of immune-mediated secondary inflammation. The presence of concomitant myocarditis in patients with amyloidosis has been demonstrated to significantly worsen the prognosis [25].

It is evident that the coexistence of myocarditis and diverse forms of cardiomyopathy is not an uncommon occurrence. However, the majority of existing literature in this field is limited to the description of individual clinical cases without any information on the etiopathogenetic management of myocarditis. A comprehensive approach to the diagnosis and treatment of myocarditis in patients with cardiomyopathies has yet to be established. The characteristics of the clinical manifestations of myocarditis and their influence on prognosis, depending on the specific type of cardiomyopathy, remain to be investigated.

## 2. Materials and Methods

**Aim:** To investigate the prevalence and clinical implications of myocarditis in individuals with cardiomyopathies and to assess the efficacy of myocarditis treatment in patients with cardiomyopathies.

Patients with primary genetically determined cardiomyopathies aged 18 years or over who provided written informed consent to participate in the study were included. The study was conducted in accordance with the ethical principles set forth in the Declaration of Helsinki and received approval from the Local Ethics Committee of Sechenov University (protocol code 10-22, dated 19 May 2022).

A total of 342 patients with primary cardiomyopathies were included in the study. The cohort comprised 125 patients with LVNC, 100 with primary myocardial hypertrophy syndrome, 70 with ARVC, 60 with DCM, and 30 with RCM (Figure 1). The study cohort also included patients with a range of mixed phenotypes: HCM + LVNC (*n* = 15), LVNC + ARVC (*n* = 9), LVNC + RCM (*n* = 6), HCM + RCM (*n* = 13). The patients were enrolled in the registry between 2008 and 2023 at an expert centre—the V.N. Vinogradov Faculty Therapeutic Clinic.

The exclusion criteria were age under 18 years, pregnancy or breastfeeding, mental retardation or incapacity, decompensated mental disorders, decompensated congenital heart disease with right heart overload, pulmonary embolism, primary pulmonary hypertension, and acquired heart valve diseases (rheumatic or due to infective endocarditis). Patients with coronary artery stenosis of 70% or more, post-infarction cardiosclerosis, left ventricular hypertrophy resulting from arterial hypertension or congenital or acquired heart valve defects, alcoholic cardiomyopathy, systemic immune diseases, oncological diseases, and sarcoidosis were also excluded.

The general clinical examination entailed the collection of patient complaints, a comprehensive history, and an objective examination. In the context of family history, particular attention was paid to the occurrence of sudden cardiac deaths in relatives, with a focus on those under the age of 35, as well as the presence of cardiomyopathies and rhythm and conduction disorders in first- and second-degree relatives. The age of disease onset, potential correlation with a preceding infection, and the possibility of an acute onset of symptoms were meticulously documented. All patients underwent a standard set of laboratory investigations, including a full blood count and biochemical panel, as well as instrumental investigations aimed at diagnosing cardiomyopathies, such as a 12-lead electrocardiogram (ECG) at rest, an EchoCG, and 24 h ECG monitoring. Additional imaging modalities, including cardiac MRI, CT, or scintigraphy and coronary angiography, were employed to substantiate the diagnosis. The frequency of these studies varied according to the specific type of cardiomyopathy, as detailed in Table 1.

In addition, medical genetics counselling was provided to patients, with DNA diagnosis subsequently offered in the majority of cases (Table 1). The isolation of DNA from the peripheral blood of patients was conducted using the phenol–chloroform deproteinisation method. The amplification of the studied DNA fragments was conducted via PCR on the Veriti (Applied Biosystems, Waltham, Massachusetts, USA) and Tertsik (DNA-Technology, Russia) amplifiers. The DNA diagnostics were conducted at different time points of this study and employed the following techniques:(1)Bidirectional direct sequencing was conducted on an ABI 3730 XL automated sequencer (Applied Biosystems).(2)High-throughput semiconductor sequencing was conducted on the PGM IonTorrent platform (Thermo Fisher Scientific, Waltham, Massachusetts, USA). A panel of genes for high-throughput sequencing was designed using AmpliSeq technology (Thermo Fisher Scientific). The presence of the mutations was confirmed by Sanger sequencing.(3)Whole exome sequencing was conducted on the NextSeq550Dx instrument (Illumina, San Diego, California, USA).

In cases 2 and 3, the presence of the identified mutations was confirmed through direct bidirectional Sanger sequencing. To ascertain the potential clinical significance, all identified genetic variants with a minor allele frequency of less than 5% as reported in the Exome Sequencing Project, 1000 Genomes, and ExAC databases were characterised using the American College of Medical Genetics (ACMG) recommendations [26], the Guidelines for Interpreting Data from Massively Parallel Sequencing Methods [27], and literature and bioinformatic data.

For DNA diagnostics, with the exception of whole exome sequencing, the following scope was applied. The following genes were analysed in the context of ARVC: *PKP2*, *DSG2*, *DSP*, *DSC2*, *JUP*, *TMEM43*, *TGFB3*, *PLN*, *LMNA*, *DES*, *CTTN3*, *EMD*, *SCN5A*, *LDB3*, *CRYAB*, and *FLNC*. The LVNC gene panel included *MYH7*, *MYBPC3*, *TAZ*, *TPM1*, *LDB3*, *MYL2*, *MYL3*, *ACTC1*, *TNNT2*, and *TNNI3B*. In the group with primary myocardial hypertrophy, a panel of sarcomeric genes, *MYBPC3*, *TAZ*, *TPM1*, *LDB3*, *MYL2*, *ACTC1*, *MYL3*, *MYH7*, *TNNI3*, *TNNT2*, and/or a targeting study of genes responsible for the occurrence of HCM phenocopies, *GLA*, *LAMP2*, *TTR*, *FXN*, *PTPN11*, etc., was employed. The DCM panel comprised the genes *ABCC9*, *ACTN2*, *ANKRD1*, *BAG3*, *CALR3*, *CAV3*, *CAVIN4*, *CSRP3*, *ILK*, *DES*, *DLG1*, *DMD*, *DTNA*, *EMD*, *EYA4*, *FHL1*, *FHOD3*, *FKTN*, *FLNA*, *FOXC1*, *FOXC2*, *GATA4*, *GATA5*, *GATA6*, *GATAD1*, *HSPB1*, *JPH2*, *LAMA4*, *LAMP2*, *LMNA*, *MIB1*, *MYBPC3*, *MYH6*, *MYH7*, *MYLK2*, *MYOZ2*, *MYPN*, *NEBL*, *NEXN*, *NKX2-5*, *NOTCH1*, *NOTCH2*, *PDLIM3*, *PLN*, *PRKAG2*, *PSEN1*, *PSEN2*, *RBM20*, *SDHA*, *SGCA*, *SGCB*, *SGCD*, *SGCE*, *SGCG*, *SMAD6*, *SNTA1*, *TCAP*, *TMPO*, *TNNC1*, *TNNT2*, *TPM1*, *TTN*, and *VCL*. In the RCM cohort, the genes that were subjected to evaluation were *DES*, *MYH7*, *TNNI3*, *TNNT2*, *ACTN1*, *FLNC*, *TTN*, *TTR*, and additional genes, contingent on the phenotypic features of each patient.

A whole exome sequencing approach was employed to investigate the genetic basis of diverse forms of cardiomyopathies. Bioinformatics searches for genetic variants were conducted within the Hereditary Heart and Vascular Diseases panel, which encompasses 302 genes (Appendix A).

The diagnosis of myocarditis was established using a combination of techniques, including myocardial morphological examination and/or a non-invasive diagnostic algorithm (Table 1).

The material for morphological investigation was obtained in the majority of cases during EMB of the right ventricle (RV) (*n* = 76). EMB was performed in accordance with the standard protocol, with access through the femoral vein using the Cordis STANDARD 5.5 F 104 FEMORAL forceps, with a sample of 3–5 myocardial fragments. In some cases, the material was obtained during open heart surgery (intraoperative myocardial biopsy, *n* = 2), explanted heart examination (*n* = 2), or autopsy (*n* = 6). A total of 86 patients underwent myocardial morphological examination.

Myocardial analysis comprised a standard histological examination conducted under a light microscope with the use of haematoxylin–eosin and Van Gieson picrofuchsin staining (for the detection of connective tissue), in addition to the following: periodic acid Schiff (PAS) staining for the detection of glycogen and other complex carbohydrates, Congo red staining for amyloid with examination of preparations in polarising light, and, in some cases, Masson and Perls staining. Additionally, an immunohistochemical study of the myocardium was conducted using antibodies (Ab) to CD3, CD20, CD45, and CD68. The myocardium was evaluated by PCR to ascertain the presence of a cardiotropic virus genome, specifically adenoviruses, herpes simplex virus type 2, cytomegalovirus, herpes simplex virus type 1, Epstein–Barr virus, varicella-zoster virus, parvovirus B19, human herpesvirus 6, human herpes virus 8, and in some cases hepatitis B and C viruses, and SARS-CoV-2 may also be present. The Dallas criteria were employed for the diagnosis of myocarditis in the myocardial morphological examination, with the additional utilisation of an immunohistochemical study with Ab to markers of T-lymphocytes (CD45+ and CD3+), macrophages (CD68+), and B-lymphocytes (CD20+) [28].

In patients who had not undergone a myocardial morphological examination, a diagnosis of myocarditis was made on the basis of a non-invasive algorithm [29]. The algorithm is validated, it is based on the assessment of the diagnostic value of various non-invasive criteria in comparison with the data of myocardial morphological examination in 100 patients [29]. The presence of a complete anamnestic triad (i.e., an association between the disease onset and infection, an acute onset, and a disease duration of less than one year), systemic immune manifestations, and high titres of anti-cardiac Ab were significant factors in the diagnosis. The Lake Louise criteria were used for the interpretation of MRI data [30]. In addition, late contrast enhancement of subepicardial localisation observed on cardiac CT and myocardial scintigraphy results was evaluated as an additional criterion.

Serum anti-cardiac Ab titres were determined by indirect immunofluorescence analysis. The Ab titres to Ag of the endothelium (AbEnd), cardiomyocyte Ag (AbCM), smooth muscle Ag (AbSM), cardiac conduction fibres Ag (AbCF), and cardiomyocyte nuclei Ag (specific anti-nuclear factor, ANF) were evaluated. To assess the Ab titres, bovine myocardial fragments were frozen in liquid nitrogen, after which slices prepared in the cryostat were incubated with patient serum at various dilutions (1:40, 1:80, 1:160, and 1:320). After incubation the slices were washed with phosphate buffer and fluoresceinisothiocyanate-labelled Ab against human IgG was applied. Subsequently, the slices were incubated once more, after which they were washed with phosphate buffer and coverslipped with 60% glycerol. The results were examined using a Leica luminescence microscope DM4000B(Wetzlar, Germany) at a magnification of ×400. The presence of fluorescent luminescence in the various structures of the bovine myocardium treated with patient serum at each dilution (1:40 to 1:320) was evaluated. The presence of anti-nuclear Ab at any titre was considered diagnostic. For the other Abs, titre values of 1:160–1:320 were found to be diagnostically significant.

**Study design**. Patients were divided into two groups according to the results of the examination: the primary group and the comparison group. The primary group consisted of patients with a combination of cardiomyopathy and myocarditis. The second group, used for comparison, consisted of patients with isolated cardiomyopathies without myocarditis. Patients in both groups were subdivided according to the type of cardiomyopathy (ARVC, HCM, DCM, LVNC, and RCM). A comparison was conducted between patients with and without myocarditis within the respective subgroups. The treatment regimen included the administration of immunosuppressive therapy (IST) to patients with myocarditis without contraindications and with the possibility of regular laboratory and instrumental follow-up. Patients from both groups were treated with cardiotropic, anti-arrhythmic, and diuretic therapies if indicated. Surgical treatment, including radiofrequency ablation of arrhythmogenic foci, implantation of cardioverter-defibrillators (ICDs), and heart transplantation, was performed if necessary. In addition, the titres of anti-cardiac Ab in the presence of myocarditis were evaluated over time, and ECG at rest, 24 h ECG monitoring, and EchoCG were repeated regardless of the presence of myocarditis. The primary endpoints were death and heart transplantation, while the secondary endpoints were progression of CHF, syncope, sustained VT, and appropriate ICD intervention.

**Statistical analysis.** The data were subjected to statistical analysis using IBM SPSS Statistics, version 26. The presentation of discrete data is in the form of distributions of absolute values and percentages. Continuous data are presented as the arithmetic mean ± standard deviation when the distribution of values is normal, or as quartiles of 50 [25; 75] when the distribution of the studied values differ significantly from normal. The normality of the distribution was evaluated using the one-sample Kolmogorov–Smirnov test when the number of observations was 50 or greater, and the Shapiro–Wilk test was employed when the number of observations was less. Differences between groups were evaluated using a Student *t*-test (for variables with normal distribution and *n* ≥ 50) or Mann–Whitney U test (for variables without normal distribution and *n* < 50). Risk factors were assessed using Cox regression in all subgroups, with the exception of RCM, which had an insufficient number of observations. The survival rates, contingent on the presence or absence of myocarditis, are illustrated graphically as Kaplan–Meier curves.

## 3. Results

Given the clinical heterogeneity of genetic cardiomyopathies, the data will be presented in a group format, according to phenotype.

### 3.1. The Primary Myocardial Hypertrophy Syndrome

Of the 100 patients in this cohort, those with HCM were the most prevalent (68%), while amyloidosis with cardiac involvement was diagnosed in 16% of cases. Additionally, less common causes of myocardial hypertrophy were confirmed, including storage diseases (10%), neuromuscular disorders (3%), myocardial hypertrophy with pronounced restrictive hemodynamics (2%), and LEOPARD syndrome (1%).

In patients with HCM, the prevalence of myocarditis was 30.9% (*n* = 21). Morphological verification of the diagnosis was achieved in 11 (52.4%) patients, with all cases exhibiting lymphocytic myocarditis (Figure 2). Additionally, 63.6% of patients (*n* = 7) demonstrated the presence of viral genomes in the myocardium, predominantly PVB19 and HHV6.

HCM patients with myocarditis had significantly elevated Ab titres against cardiomyocyte Ag compared to HCM patients without myocarditis (Figure 3).

Patients with myocarditis exhibited more severe clinical manifestations (Table 2), including a significantly higher incidence of LVEF decline and a higher CHF functional class (NYHA). Conversely, they exhibited a twofold reduction in the incidence of atrial fibrillation, which suggests that the main cause of decompensation was myocarditis rather than the progression of the HCM.

Furthermore, patients with myocarditis exhibited a significantly elevated incidence of fatal outcomes. In univariate analysis, the presence of myocarditis was identified as an independent predictor of mortality (HR 6.0, 95% CI 1.24–29.07, *p* = 0.026; Figure 4).

It is noteworthy that pathogenic or likely pathogenic variants in the myosin heavy chain gene were more frequently identified among patients with myocarditis than in the group without myocarditis (Figure 5).

In 48% of cases, patients with HCM and myocarditis received IST. In approximately 50% of cases, methylprednisolone was administered at a mean dose of 24 mg/day (range [14; 40] mg/day). This was provided in combination with mycophenolate mofetil (2 g/day) in one patient. Three patients received hydroxychloroquine (200–400 mg/day), while two patients were treated with azathioprine (100–200 mg/day). Three patients with positive myocardial virus results received anti-viral therapy, with two receiving acyclovir and one receiving valacyclovir. Patients who received IST exhibited a favourable response in terms of suppression of premature ventricular contractions (PVCs) (1122 [164; 10,000] → 1025 [85; 3138], *p* = 0.043) and a reduction in anti-cardiac Ab titres compared to patients with HCM and myocarditis who did not receive IST. EchoCG parameters remained stable (*p* > 0.05).

In other forms of primary myocardial hypertrophy syndrome, amyloidosis with cardiac involvement, storage diseases, and neuromuscular disorders, myocarditis was diagnosed in approximately one-third of cases (Figure 6). In some subgroups, such as AL amyloidosis and Fabry disease, the incidence of myocarditis reached 50%. The presence of myocarditis also had a deleterious effect on the course of the disease.

### 3.2. Left Ventricular Non-Compaction

In LVNC, the prevalence of myocarditis was 54.4% (*n* = 68). Morphological verification of the diagnosis was performed in 21 (30.9%) cases. In all patients the myocarditis was lymphocytic (Figure 2). In addition, the viral genome was present in 47.6% of cases (*n* = 10), the most common being PVB19 (*n* = 6), followed by a combination of HSV1 and HHHV6 (*n* = 1), EBV (*n* = 1), a combination of HHV6 and EBV (*n* = 1), and a mixture of HHV6, EBV, CMV, and PVB19 (*n* = 1). In patients with myocarditis, there was a notable elevation in the titres of anti-cardiac Ab directed against Ag expressed by cardiomyocytes, smooth muscle cells, and fibres of the conducting system (Figure 3).

Patients with myocarditis exhibited more severe clinical manifestations across a range of parameters, including systolic dysfunction, CHF functional class, and higher prevalence of ventricular rhythm disturbances that necessitated ICD implantation and resulted in appropriate shocks (Table 3).

Despite appropriate treatment of CHF and rhythm disturbances, patients with myocarditis were more likely to experience adverse outcomes. The presence of myocarditis was identified as an independent predictor of mortality (HR 5.8, 95% CI 1.3–25.0, *p* = 0.02; Figure 4). Among patients with myocarditis, pathogenic or likely pathogenic variants in the *desmoplakin* and *desmin* genes were of particular interest, as they distinguished this group from patients without myocarditis (Figure 5).

A total of 63% of patients with a diagnosis of myocarditis received IST. Twenty-two patients were prescribed glucocorticoids, with a mean starting methylprednisolone dose of 24 [16; 32] mg/day. Azathioprine was administered to nine patients, with a mean dose of 150 [75; 175] mg/day. Hydroxychloroquine was administered in 23 cases, with a daily dose of 400 mg in five cases and 200 mg in the remaining eighteen cases. It should be noted that combinations of drugs were administered, as well as switching from one regimen to another. Consequently, eleven patients were treated with glucocorticoid monotherapy, two patients received azathioprine, and nineteen patients received hydroxychloroquine. A combination of glucocorticoids and azathioprine was employed in seven patients, while four patients initially received glucocorticoids and were subsequently transitioned to a maintenance IST with hydroxychloroquine. There was a positive immunological dynamic with respect to the titres of anti-cardiac Ab to the Ag of the fibres of the conducting system, with a shift from 1:160 [1:160; 1:320] to 1:160 [1:80; 1:320], *p* = 0.003, vs. 1:160 [1:80; 1:320] to 1:80 [1:80; 1:160], *p* = 0.257, in LVNC patients with myocarditis without IST. Significant LVEF improvement (36.8 ± 13.3 → 40.3 ± 12.4, p = 0.008 vs. 31.4 ± 11.8 → 35.6 ± 13.6, *p* = 0.195) and a significantly lower incidence of reaching the combined endpoint of death or heart transplantation—20.9% vs. 44.0%, *p* = 0.042—were observed in patients with LVNC and myocarditis who received IST in comparison with the same patients without IST.

### 3.3. Arrhythmogenic Right Ventricular Cardiomyopathy

In the ARVC cohort all 70 patients were diagnosed with the classical “dominant-right” phenotype. The vast majority of patients had a definite diagnosis (*n* = 59) and the rest had a borderline diagnosis (*n* = 11). The incidence of myocarditis in the ARVC group was 74.3% (*n* = 52). Morphologically, lymphocytic myocarditis (Figure 2) was confirmed in 15.4% (*n* = 8) of cases, and it was virally positive in half of the patients in this subgroup (*n* = 4: 1 SARS-CoV-2, 1 PVB19, 2 HHV6, including 1 in combination with HSV1). It is notable that the myocarditis patients exhibited elevated immunological activity, as evidenced by significantly higher Ab titres against four out of the five cardiac Ags (Figure 3).

Patients with myocarditis exhibited lower LVEF and an elevated CHF incidence (Table 4). In patients with myocarditis, the efficacy of radiofrequency ablation of ventricular arrhythmias was found to be approximately half that observed in comparable cases in patients without myocardial inflammation. Conversely, patients with myocarditis exhibited a reduced incidence of sustained VT, RV fat replacement, and thinning, which are typical characteristics of ARVC progression. This suggests that myocarditis, rather than progression of the fibro-fatty replacement in frames of ARVC, led directly to decompensation of the disease. In ARVC, the presence of myocarditis did not significantly impact the incidence of lethal outcomes (Figure 4).

The predominant pathogenic or likely pathogenic variants in patients with myocarditis were observed in the *desmoplakin* and *filamin C* genes, in contrast to patients with isolated ARVC (Figure 5).

IST was administered to 73% of patients who had combination therapy with ARVC and myocarditis. The mean dose of glucocorticoids was 20 [8; 24] mg/day. Five (13.2%) patients received azathioprine at a dose of 100 [0; 125] mg/day, and thirty (79.0%) patients received hydroxychloroquine 200 mg/day. The administration of IST was associated with a reduction in the incidence of fatal outcomes (5.3% vs. 28.6%, *p* = 0.021). Additionally, LVEF remained stable, whereas in the group without IST, there was a trend towards its progressive decrease and left-sided dilatation (Table 5). Furthermore, a positive response was observed for all Ab titres directed against the heart muscle in the presence of IST.

### 3.4. Primary Dilated Cardiomyopathy

In the DCM cohort, the incidence of myocarditis was 56.7% (*n* = 34), including a morphologically verified diagnosis in 47.1% (*n* = 16) of patients. In all 16 cases, the presence of lymphocytic myocarditis was confirmed (Figure 2). Additionally, viral genome detection was observed in myocardium in 25% of patients (*n* = 4). The PVB19 genome was identified in a single patient. A single patient exhibited a combination of PVB19 and HHV6. One patient exhibited evidence of infection with EBV, while another demonstrated the presence of four distinct viruses, namely EBV, HHV6, CMV, and PVB19, simultaneously. Patients with myocarditis exhibited significantly elevated titres of anti-cardiac Ab to fibres of the conduction system, accompanied by significantly reduced LV EF and thinner LV walls (Table 6). The patients were more in need of administration of loop diuretics, particularly thorasemide. In patients with myocarditis, non-sustained VT requiring amiodarone and ICD implantation was registered significantly more often. All endpoints were reached more often in the myocarditis subgroup of patients.

The presence of myocarditis diagnosed by any method (morphologically or non-invasively) was not identified as a predictor of mortality. However, the presence of morphologically verified myocarditis was identified as a lethal outcome predictor (HR 3.6, 95% CI 1.433–9.249, *p* = 0.004).

Among patients with myocarditis, pathogenic or likely pathogenic variants in the *desmin* gene were more prevalent (Figure 5). These mutations were not observed among patients without myocarditis and may contribute to a favourable background for the onset of inflammation.

A total of 62% of patients with DCM received IST: 76.2% (*n* = 16) of patients received glucocorticoids, with a mean dose of 24 [16; 30] mg/day, additionally, 42.9% (*n* = 9) of patients were administered hydroxychloroquine at a dose of 200 mg/day, while 23.8% (*n* = 5) were treated with azathioprine at a dose of 150 [62.5; 150] mg/day. In the group receiving IST, there was a clear trend towards lower mortality (23.8 vs. 53.8, *p* = 0.080) and a significant improvement in LVEF (29.8 ± 11.7 → 35.9 ± 13.9, *p* = 0.036 vs. 26.5 ± 11.9 → 33.6 ± 14.2, *p* = 0.208) compared to DCM patients with myocarditis who did not receive IST. Furthermore, the patients exhibited a positive immunological response, as evidenced by anti-cardiac Ab level decline, including those of the endothelium, cardiomyocytes, and conducting system fibres.

### 3.5. Restrictive Cardiomyopathy

In primary RCM, myocarditis was identified in one-third of cases (*n* = 5), of which in two patients (40%) the diagnosis was confirmed morphologically (by EMB) (lymphocytic myocarditis, Figure 2). In one case, myocarditis was confirmed to be positive for the SARS-CoV-2 virus. Patients with myocardial inflammation exhibited higher Ab titres to Ag of cardiomyocyte nuclei, endothelium, and cardiomyocytes (Figure 3), as well as worse atrioventricular conduction, more frequent PVCs, larger volumes of both atria, and more pronounced restrictive diastolic dysfunction (Table 7).

The incidence of fatal outcomes was not found to be significantly influenced by the presence of myocarditis. Among the patients with positive results of DNA diagnostics (40.0%), pathogenic or likely pathogenic variants in the myosin-binding protein gene were the most prevalent in myocarditis patients (Figure 5).

Three out of the five patients diagnosed with myocarditis were administered IST. In two cases, the initial dosage of methylprednisolone was 16 mg/day, which included a combination with hydroxychloroquine 200 mg/day in one patient. In the third case, a less aggressive IST was performed, comprising an initial dose of 8 mg/day methylprednisolone and 200 mg/day hydroxychloroquine. During the follow-up of patients with myocarditis, a threefold decrease in the incidence of ventricular ectopy (5043 [827; 12,419] → 145 [72; 351], *p* = 0.068), a reduction in the frequency of VT (60.0% → 20.0%, *p* = 0.016), and a trend towards a decrease in atrial size and severity of diastolic dysfunction were observed.

Furthermore, the incidence of myocarditis in other causes of restrictive phenotype was also evaluated. In the case of amyloidosis with cardiac involvement (*n* = 13), the incidence of myocarditis was found to be 44%. Morphological verification of myocarditis was conducted in a single case of Danon’s disease (*n* = 1), while in a case of Loeffler’s pancarditis (*n* = 1) it was diagnosed clinically. The presence of myocarditis deteriorated the manifestations of CHF and rhythm disturbances.

The incidence and role of myocarditis were evaluated in patients presenting with mixed phenotypes as well. In patients with HCM + LVNC, the incidence was 40.0%, in ARVC + LVNC it was 88.9%, and in RCM + LVNC it was 42.9%. In all cases, the presence of myocarditis resulted in the onset or worsening of CHF and an exacerbation of rhythm disturbances. It should be noted that in HCM alone, myocarditis was diagnosed in 30.9%, in LVNC in 54.4%, in ARVC in 74.3%, and in RCM in 37.5%. Thus, a mixed phenotype with LVNC increases the predisposition to myocarditis in other cardiomyopathies. Overall, patients with mixed phenotypes combined with myocarditis had the most severe disease course compared to patients with mixed phenotypes without myocarditis or compared to patients with a pure phenotype combined with myocarditis.

## 4. Discussion

Myocarditis is a common phenomenon in cardiomyopathies, occurring either as a primary viral process or as a secondary autoimmune reaction, as indicated by elevated Ab titres directed against cardiac Ag. The prevalence of myocarditis differed according to the specific genetic form of cardiomyopathy (Figure 7).

In the vast majority of cases, we detected lymphocytic myocarditis with variable numbers of macrophages (CD68+, more often in post-COVID-19 myocarditis). Lymphocytes were represented by T-lymphocytes (CD45+ and CD3+). In the immunohistochemical study with antibodies against CD, we almost never detected B-lymphocytes (CD20+). Apoptosis was detected together with other variants of cell death (necrosis). The presence of apoptosis and necrosis was associated with more pronounced lymphocytic infiltration and indicated high myocarditis activity.

According to our data, the incidence of myocarditis did not differ between severe and mild phenotypes. Rather, the role of myocarditis in the development of the clinical picture was different: in patients with mild genetic cardiomyopathy, myocarditis became a kind of trigger for the onset of symptoms, and often the diagnosis of genetic cardiomyopathy was made precisely at the stage of myocarditis development. In patients with initially more aggressive phenotypes, the development of myocarditis mainly led to severe decompensation in previously stable patients. In general, myocarditis in genetic cardiomyopathies is a contributing factor to the activation (in mild phenotypes) and realisation (in more aggressive phenotypes) of an abnormal genetic programme.

The highest incidence was observed in ARVC. Myocarditis was observed in almost three-quarters of patients, and the significant contribution of autoimmune aggression to the development of inflammation was noteworthy. This is in contrast to other cardiomyopathies, in which not all types of anti-cardiac Ab are elevated concurrently. In addition to the immune mechanism, a primary viral component was also identified, as evidenced by the detection of a viral genome in myocardial cells in half of the patients. It is also noteworthy that in approximately half of the patients with ARVC, the disease manifested acutely and was linked to a preceding viral infection. The frequency of pathogenic mutation detection was also the highest in ARVC, amounting to 25%. The contribution of inflammation to the development and progression of ARVC has been extensively discussed in the literature, with reference to both “trivial” viral myocarditis [31,32] and auto-Ab [33,34,35]. The mortality rate in patients with a combination of myocarditis and ARVC was the lowest, which can be attributed to the favourable outcomes observed in those with myocarditis, due to the potential for influencing the autoimmune component with IST. Therefore, myocarditis in ARVC is a significant contributor to the formation of the disease phenotype and the disease’s overall outcome.

The second most frequently observed concomitant myocarditis was in primary DCM, accounting for 56.7% of cases. It is noteworthy that the viral genome was detected in only a quarter of the patients, a frequency about half that observed in ARVC, HCM, or LVNC. This may indicate a greater role for immune mechanisms in the development of myocarditis in this cardiomyopathy. It is of particular note that AbEnd is increased in comparison to patients with isolated DCM, which is absent in cardiomyopathies other than ARVC. This may be indicative of vasculitis against the background of myocarditis. The frequency of pathogenic or likely pathogenic variant detection in patients with the combination of primary DCM and myocarditis was comparable with other groups, amounting to 20%. The presence of mutations that result in the synthesis of altered cardiomyocyte proteins may also represent an additional target for the development of autoimmune myocardial damage [2]. Despite the relatively limited number of patients with a positive viral result, the vast majority of patients (91.2%) presented with an acute onset of the disease, a phenomenon that is considerably less prevalent in isolated cardiomyopathies. Furthermore, in approximately half of these cases, the onset was associated with a previous infection. It seems probable that the trigger for the development of inflammation in these cases was not the virus itself and its inherent properties, but rather the pathological activation of the immune system subsequent to the disease [36]. The mortality rate among patients with DCM combined with myocarditis was the highest (35.3%) in comparison to other cardiomyopathies. This can be seen as a manifestation of the underlying disease, but it also reflects the severity of myocarditis itself. The mortality rate was lower (26.9%) among patients with DCM without myocarditis, and it was even higher (53.8%) among patients with myocarditis who did not receive IST.

In terms of frequency, LVNC occupies the third position, with a slight margin separating it from DCM (54.4%). The non-compact layer is a favourable target for viral attachment [18,19,20], as evidenced by the fact that almost half of the patients were virus-positive. Furthermore, acute onset of the disease was recorded in 75% of cases. Concurrently, an active immune component was also present, as evidenced by elevated Ab titres for AbCM, AbSM, and AbCF. The frequency of pathogenic or likely pathogenic variant detection in patients with a combination of LVNC and myocarditis was the lowest in comparison to other cardiomyopathies. This is attributable to the high genetic heterogeneity of LVNC and the fact that not all genes that may lead to a non-compaction phenotype have been identified. The incidence of unfavourable outcomes in the combination of myocarditis with LVNC was relatively high (20.6%), with a further 7.9% having undergone heart transplantation. In the absence of this procedure, the outcome would also have been unfavourable. As in DCM, this is not only a feature of the cardiomyopathy itself but also the influence of myocarditis. In patients with myocarditis who did not receive IST, these figures are significantly higher (death, 36%; death+ heart transplantation, 44%) compared to patients with isolated LVNC, in whom mortality was only 5.3%.

The fourth highest incidence of myocarditis was observed in the RCM group (37.5%). The frequency of viral genome detection in this group was relatively low in comparison to other cardiomyopathies, as well as in DCM. However, this may be attributed to the limited number of patients who underwent myocardial morphological examination and the fact that this group is the smallest due to the rarity of RCM. Nevertheless, the acute onset and association with infection in patients with myocarditis combined with RCM were also recorded less frequently than in other cardiomyopathies. Furthermore, there were no differences in these features between patients with and without RCM combined with myocarditis. This may indicate that the course of myocarditis in RCM is more hidden and latent. A distinctive feature of this group was a significant increase in ANF titre compared with other cardiomyopathies, while other Abs remained relatively low and did not differ statistically significantly from those in patients with isolated RCM. This may reflect a specific mechanism of myocarditis in RCM that requires further investigation. The mortality rate associated with the coexistence of myocarditis and RCM was found to be minimal and comparable to that observed in ARVC. As for treatment results, further study on a larger number of patients is required to gain a full understanding of the results of myocarditis treatment in patients with RCM.

Concomitant myocarditis was observed with the lowest frequency in HCM. The frequency of this occurrence was 30.9%, with HCM exhibiting the highest percentage of virus-positive myocarditis (58.3%). This indicates that hypertrophied myocardium, as well as non-compacted layer, may be a susceptible target for viral infection, particularly given that both HCM and LVNC may be attributable to mutations in sarcomeric genes, which emphasises their common characteristics. With regard to the frequency of acute onset and the association of onset with infection, HCM is only slightly less prevalent than LVNC. Nevertheless, we found no data in the literature on the molecular mechanisms that might favour adhesion or rapid spread of the virus to the myocardium due to pathogenic or likely pathogenic variants in sarcomeric genes, as in genes associated with DCM. With regard to the frequency of detection of pathogenic mutations, HCM is ranked second only to ARVC and RCM, with a prevalence of one-fifth among patients with myocarditis. A distinctive feature of HCM in conjunction with myocarditis is the occurrence of an isolated elevation in Ab titres directed towards cardiomyocytes, in comparison to patients exhibiting isolated HCM. This may be indicative of the fact that cardiomyocytes are the direct target of an autoimmune attack, including the disruption of the structure of sarcomeric proteins. A review of the literature reveals a paucity of studies examining anti-cardiac Ab in HCM. The majority of these studies, dating from the 1970s to 1980s, merely document the presence of such Abs in patients with idiopathic hypertrophic subaortic stenosis. The most recent study, published in 1992, offers a more comprehensive examination of this topic. B. Maisch demonstrated that 78% of patients with HCM had Ab to sarcolemma and 43% had Ab to myofibrils, which is consistent with our data on Ab to cardiomyocytes [37]. The incidence of fatal outcomes in patients with myocarditis and HCM was high (26.7%), ranking second only to DCM. In patients with HCM without myocarditis, the mortality rate was significantly lower, at 6.4%. Therefore, myocarditis has a considerable effect on the prognosis of HCM, as is the case with other cardiomyopathies.

One of the principal findings of our study is the beneficial impact of IST for myocarditis. While there are several papers in the literature on the combination of myocarditis with various genetic cardiomyopathies, as referenced in the Introduction, they are typically limited to a mere statement of the presence of myocarditis in this group of patients. There is a paucity of data regarding the treatment of myocarditis in genetic cardiomyopathies. Our study demonstrates that patients with cardiomyopathies benefit from administration of IST, which enables at least stabilisation (and in some cases improvement) of LV systolic function, enhances the efficacy of anti-arrhythmic therapy, and reduces the probability of adverse outcomes. Conversely, patients with a combination of myocarditis and cardiomyopathies who did not receive IST demonstrated unfavourable outcomes and a higher incidence of adverse events (in LVNC, ARVC, and DCM groups).

## 5. Conclusions

Myocarditis represents the most significant epigenetic factor in the onset and progression of genetic cardiomyopathies, exhibiting a primary, viral (including SARS-CoV-2) or secondary, autoimmune nature. Myocarditis is a prevalent phenomenon in various genetic cardiomyopathies, with an incidence ranging from 30.9% to 74.3%. The principal clinical signs of myocarditis in patients with cardiomyopathies are CHF decompensation and the emergence or worsening of rhythm disturbances. In such instances, it is imperative to conduct a comprehensive diagnosis of myocarditis. If myocarditis is confirmed, it is reasonable to administer IST, as the treatment of myocarditis in patients with cardiomyopathies can stabilise and in some cases improve the patient’s condition and reduce the likelihood of adverse outcomes. Therefore, in cardiomyopathies, the genetic basis and inflammation are components of a single continuum, which forms a complex phenotype.

## 6. Study Limitations

One of the limitations of the study is the non-invasive diagnosis of myocarditis, which was used in a number of cases since the gold standard for diagnosing this disease is morphological examination of the myocardium using EMB. Nevertheless, recent years have seen a growing reliance on non-invasive diagnosis of myocarditis in the context of medical research. Verification of the diagnosis of myocarditis by morphological examination of the myocardium was carried out in only 38.2% of cases (*n* = 222) in the European Registry of Myocarditis and Cardiomyopathies (CMY-LT Registry, EORP). In the remaining cases, the diagnosis was established on the basis of clinical criteria and/or by using MRI [38]. In our study, the diagnosis of myocarditis was not based on individual tests, but rather on the application of a comprehensive non-invasive diagnostic algorithm [29]. This approach has been shown to significantly reduce the likelihood of false-positive diagnoses.

A further limitation of the study is the necessity to compare small subgroups with one another. Despite the relatively large number of patients included in the study (*n* = 387), the patients were initially divided into five groups according to phenotype (HCM, LVNC, ARVC, DCM, RCM). Subsequently, patients within each group were compared based on the presence or absence of myocarditis. Finally, among patients with myocarditis, patients with and without IST were compared. The analysis of the patient cohort with RCM presented a considerable challenge. RCM is the rarest of cardiomyopathies, thus the group of 15 patients is of significant interest [39]. It is for this reason that the study was expanded to include this small group. However, a comparative analysis of patients with RCM with and without myocarditis is significantly limited, and the construction of prognostic models by analogy with other cardiomyopathies is not possible due to the small number of observations. Comparisons between groups with a limited number of observations may be susceptible to type II error. To reduce the probability of this error and to minimise potential bias, only suitable non-parametric statistical techniques were employed in the analysis of small groups. In addition to investigating statistically significant differences, attention was paid to a comprehensive examination of trends.

## Figures and Tables

**Figure 1 genes-16-00051-f001:**
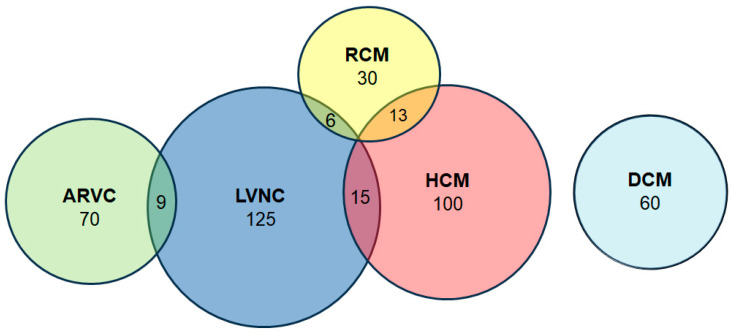
Graphical representation depicting the structure of the patients included in the study, taking into account the presence of mixed phenotypes.

**Figure 2 genes-16-00051-f002:**
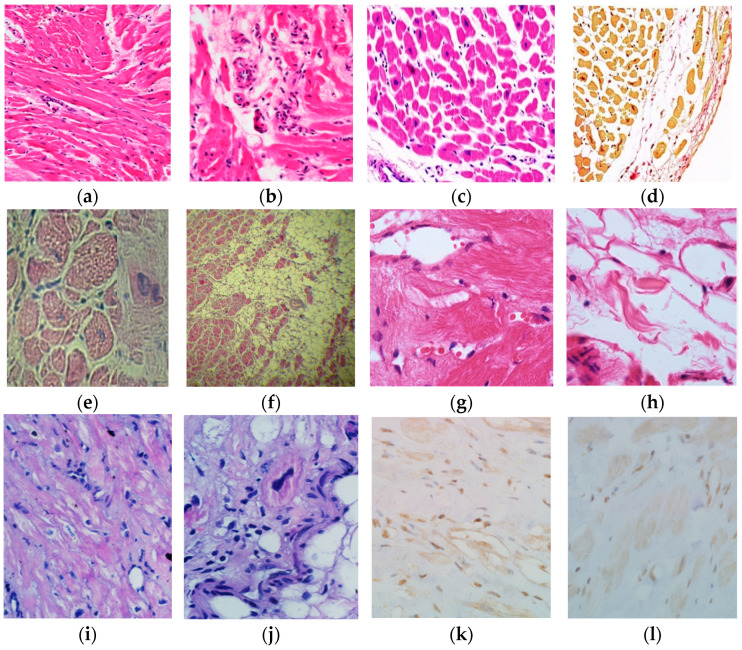
Results of morphological study of myocardium in different cardiomyopathies. (**a**,**b**) Myocardial changes in HCM in the form of bizarre shape of branching cardiomyocytes, small focal cardiosclerosis with neoangiogenesis, and lymphohistiocytic infiltrates in sclerosis foci; (**c**,**d**) myocarditis in HCM with productive capillarites and development of interstitial sclerosis: myocardium is divided by fibrous septa of unequal thickness into lobules, uneven hypertrophy of nuclei, vessels with swollen endothelium, and perivascular accumulations of lymphoid elements are noted, more than 14 in the field of view at high magnification; (**e**,**f**) picture of lymphohistiocytic infiltration in ARVC, pronounced total fibrous-fatty replacement of myocardium of LV, the area of preserved myocardium in some areas does not exceed 25%; (**g**,**h**) lymphohistiocytic infiltrates perivascularly and in the interstitium (**g**) in a patient with DCM within laminopathy, fatty tissue replacement of dead cardiomyocytes (**h**); (**i**,**j**) SARS-CoV-2-induced myocarditis in a patient with RCM caused by pathogenic or likely pathogenic variants in *MyBPC3* and *LZTR1* genes: marked lymphohistiocytic infiltration, areas of lipomatosis, dystrophic changes in cardiomyocytes; (**k**,**l**) Ab to SARS-CoV-2 nucleocapsid (**k**) and spike antigen (**l**). (**a**–**c**,**e**–**j**)—haematoxylin and eosin staining; (**d**)—Van Gieson picrofuchsin staining; (**k**,**l**)—immunohistochemical study; (**a**,**f**)—low magnification; (**b**–**e**,**g**–**l**)—high magnification.

**Figure 3 genes-16-00051-f003:**
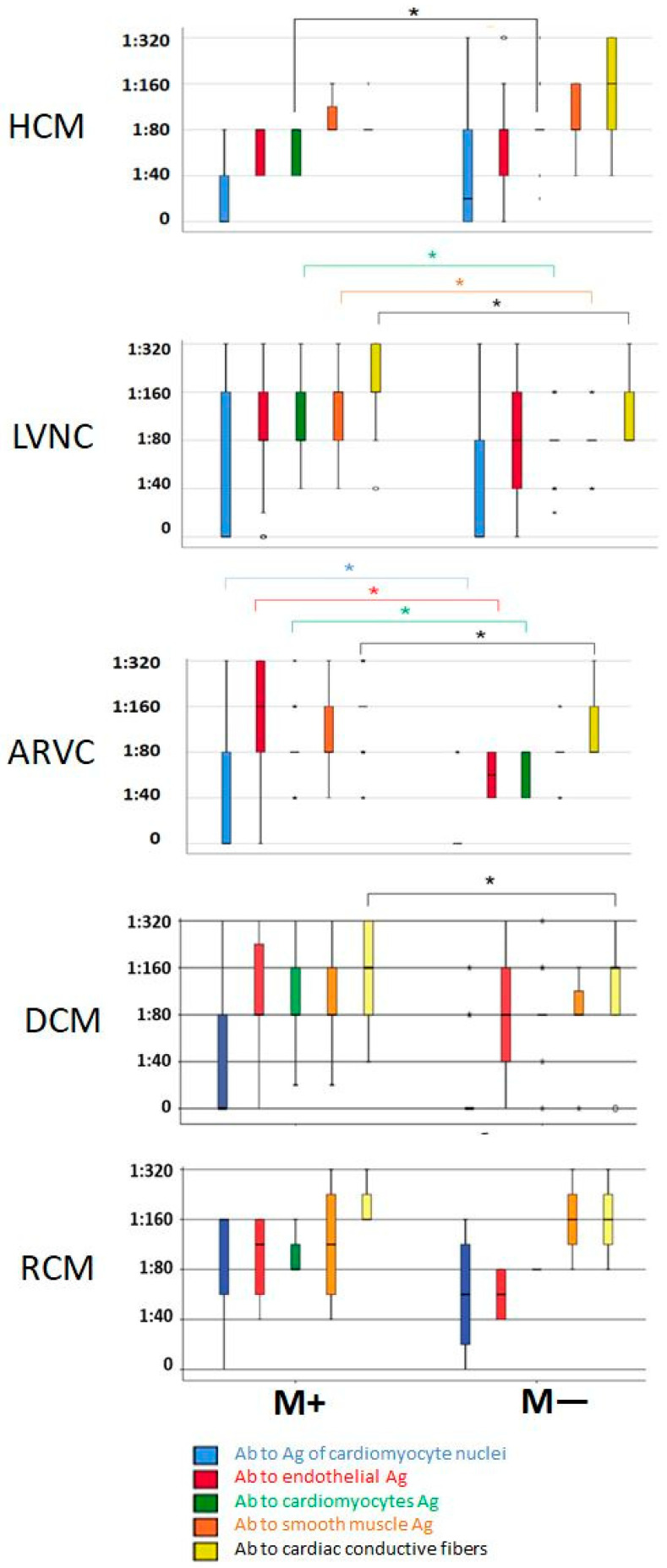
Titres of anti-cardiac antibodies in different cardiomyopathies, depending on the presence (M+) or absence (M−) of myocarditis; * - *p* < 0.05.

**Figure 4 genes-16-00051-f004:**
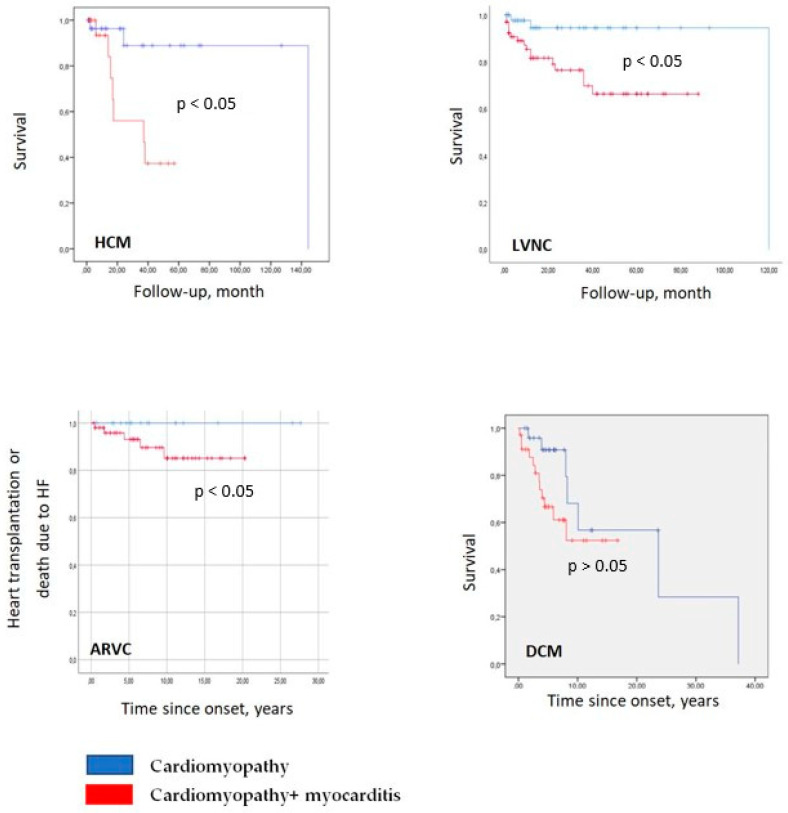
Kaplan–Meier curves for different genetic cardiomyopathies, depending on the presence or absence of myocarditis. Red colour—patients with a combination of cardiomyopathy and myocarditis, blue colour—patients with isolated cardiomyopathies.

**Figure 5 genes-16-00051-f005:**
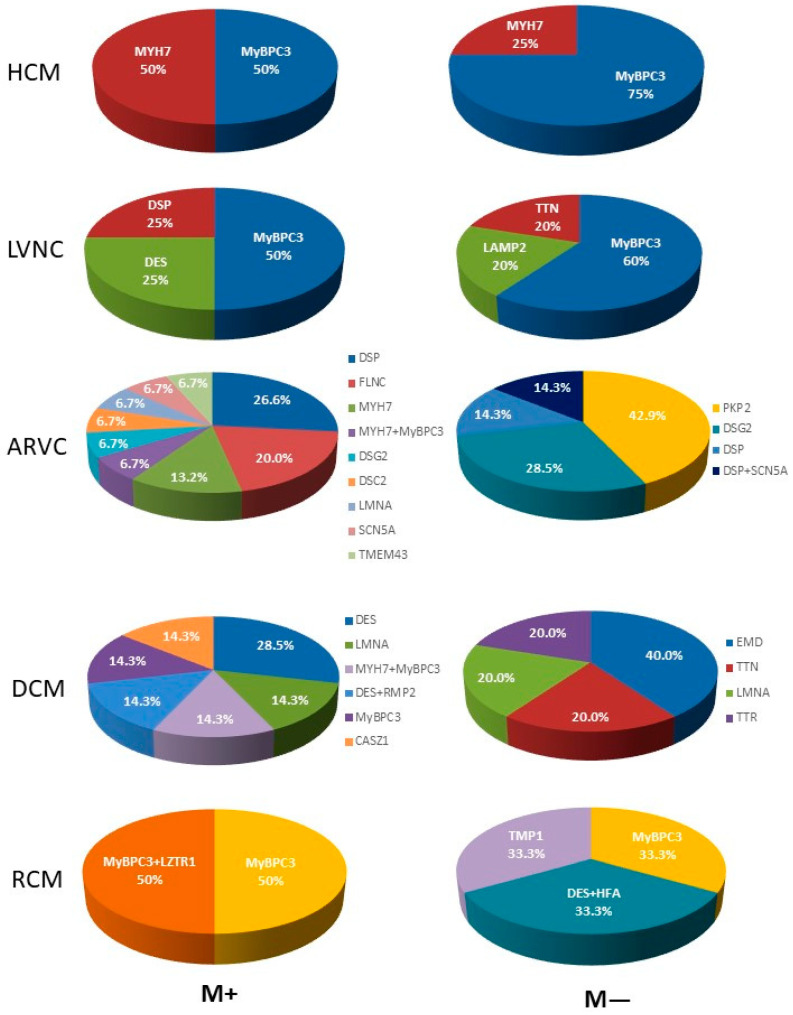
Spectrum of genes with pathogenic or likely pathogenic variants in different cardiomyopathies, depending on the presence (M+) or absence (M−) of myocarditis.

**Figure 6 genes-16-00051-f006:**
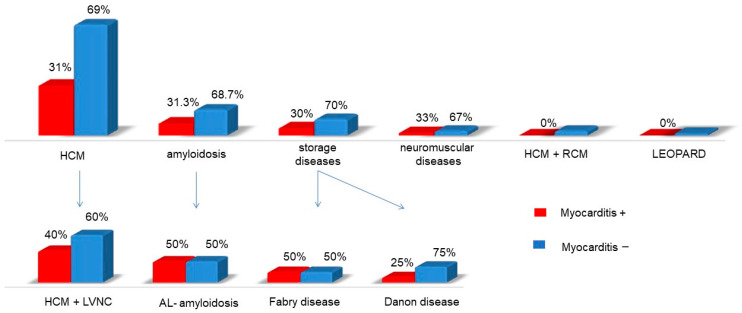
Frequency of myocarditis in different causes of myocardial hypertrophy syndrome.

**Figure 7 genes-16-00051-f007:**
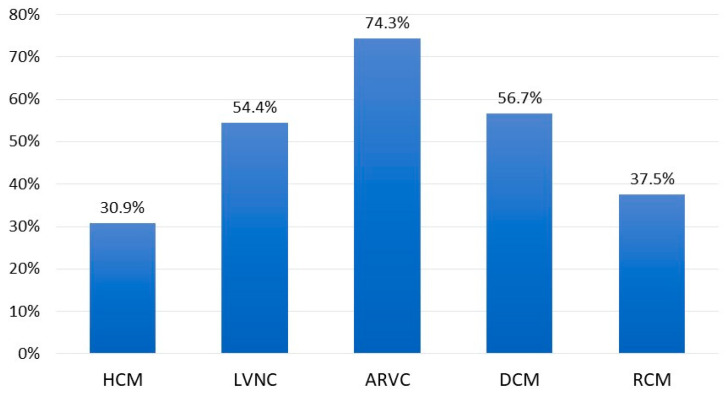
Frequency of superimposed myocarditis, depending on the type of cardiomyopathy.

**Table 1 genes-16-00051-t001:** Frequency of investigations in different cardiomyopathies.

Investigation	HCM	LNVC	ARVC	DCM	RCM
Determination of anti-cardiac Ab titres in blood serum, %	43.0	81.6	85.7	86.7	46.9
Morphological investigation of myocardium with determination of cardiotropic virus genome, %	30.0	20.8	12.9	33.3	36.7
DNA diagnostic, %	96.0	41.0	100.0	71.7	59.4
Cardiac MRI, %	31.0	84.0	91.4	30.0	37.5
Cardiac CT, %	32.0	68.0	27.1	55.0	12.5
Myocardial scintigraphy, %	11.0	21.6	7.1	18.3	18.8
Coronary angiography, %	18.0	27.2	21.4	43.3	21.9

**Table 2 genes-16-00051-t002:** Baseline characteristics and endpoints of patients with HCM, classified according to the presence or absence of myocarditis.

Characteristic	Myocarditis+	Myocarditis−	*p*
N (%)	21 (30.9)	47 (69.1)	-
Age, years	44.6 ± 12.9	48.8 ± 14.5	>0.05
Acute onset, n (%)	13 (61.9)	2 (4.3)	<0.001
Relation to prior infection, n (%)	13 (61.9)	1 (2.1)	<0.001
Myocardial morphological investigation, n (%)	11 (52.4)	9 (19.1)	0.009
Viral genome in myocardium, n (% of patients with myocardial morphological examination)	7 (63.6)	5 (55.6)	>0.05
AbCM, titre	1:80 [1:80; 1:80–1:160]	1:80 [1:40; 1:80]	0.017
Pathogenic/likely pathogenic variants, n (%)	6 (28.6)	8 (17.0)	>0.05
LV EF (EchoCG), %	51.9 ± 16.3	57.8 ± 11.2	>0.05
LV EF (EchoCG) ≤ 45%, n (%)	8 (38.1)	7 (14.9)	0.032
RV hypertrophy, n (%)	4 (19)	2 (4.3)	0.059
Maximum LV wall thickness (MSCT), mm	18.0 ± 3.0	23.1 ± 6.5	0.038
NYHA CHF class	3 [2; 3]	2 [1; 3]	0.026
Presence of atrial fibrillation, n (%)	6 (28.6)	25 (53.2)	0.044
PVCs per day, pcs.	289 [14; 3513]	63 [10; 404]	>0.05
Presence of VT, n (%)	11 (52.4)	23 (48.9)	>0.05
ICD implantation, n (%)	6 (28.6)	11 (23.4)	>0.05
Death, n (%)	7 (33.3)	3 (6.4)	0.01

**Table 3 genes-16-00051-t003:** Baseline characteristics and endpoints of patients with LVNC depending on the presence or absence of myocarditis.

Characteristic	Myocarditis+	Myocarditis−	*p*
N (%)	68 (54.4)	57 (45.6)	-
Age, years	46.3 ± 16.5	48.2 ± 16.4	>0.05
Duration of the disease, months	27 [4; 84]	53 [13; 144]	0.016
Acute onset, n (%)	51 (76)	15 (26.3)	<0.001
Relation to prior infection, n (%)	40 (58.8)	9 (15.8)	<0.001
Myocardial morphological investigation, n (%)	21 (30.9)	5 (8.8)	>0.05
Pathogenic/likely pathogenic variants, n (%)	8 (11.8)	5 (8.8)	>0.05
Family burden	9 (13.2)	17 (29.8)	0.02
Structural and functional parameters in echocardiography and MRI			
EDV	168 [120; 202]	134 [100; 182]	0.043
ESV	100 [70; 137]	81 [49; 121]	0.03
LV EF (EchoCG), %	35 ± 13	43±14	0.002
LV EF (EchoCG) ≤ 35%, n (%)	37 (54.4)	17 (29.8)	0.004
RV, ml	73 [49; 90]	54 [45; 86]	0.042
E/A	1.6 [1.1; 2.6]	1.3 [0.7; 1.8]	0.046
LGE at MRI, n (%)	27 (39.7)	9 (15.8)	0.004
Parameters characterising heart failure			
NYHA CHF class	2-3 [1; 3]	2 [1; 3]	0.029
Administration of ACE, n (%)	51 (75)	25 (43.9)	<0.001
Administration of β-blocker + ACE, n (%)	43 (63.2)	26 (45.6)	0.017
Administration of spironolactone, n (%)	47 (69.1)	26 (45.6)	0.003
Parameters characterising rhythm disturbances			
Need for anti-coagulant prescriptions, n (%)	38 (55.9)	22 (38.6)	0.017
Presence of VT, n (%)	46 (67.6)	20 (35.1)	<0.001
Sustained VT, n (%)	12 (17.6)	6 (10.5)	>0.05
Implantation of ICDs, n (%)	26 (38.2)	13 (22.8)	0.048
End points			
Appropriate ICD interventions, n (% of ICD patients)	11 (42.3)	3 (23.1)	0.181
Death, n (%)	16 (23.5)	3 (5.3)	0.004
Heart transplantation, n (%)	6 (8.8)	1 (1.8)	0.09

**Table 4 genes-16-00051-t004:** Baseline characteristics and endpoints of patients with ARVC depending on the presence or absence of myocarditis.

Characteristic	Myocarditis+	Myocarditis−	*p*
N	52	18	-
Acute onset, n (%)	55.8	44.4	0.364
Relation to prior infection, n (%)	53.8	11.1	0.002
Myocardial morphological investigation, n (%)	15.4	5.5	>0.05
Pathogenic/likely pathogenic variants, n (%)	28.9	38.9	0.380
LV EF (EchoCG), %	53.1 ± 14.3	60.0 ± 7.5	0.071
LV EF (EchoCG) ≤ 35%, n (%)	11.5	0	0.192
NYHA CHF class	15.4	0	0.09
PVC, thousands	18.2 [3.2; 36.0]	13.7 [2.1; 19.0]	0.161
RFA, %	30.8	16.7	0.187
Efficiency of RFA, %	50	100	0.470
Sustained VT, %	26.9	50.0	0.075
Fat in the RV (MRI), %	42.3	66.7	0.040
RV thinning, %	30.8	61.1	0.010
Death, %	11.5	16.7	0.420
Death/HT due to progressive CHF	9.6	0	0.215

**Table 5 genes-16-00051-t005:** Follow-up of patients with ARVC combined with myocarditis who did and did not receive IST.

Characteristic	Baseline	Follow-Up	*p*
IST+			
LV EF, %	51.0 ± 13.6	50.9 ±12.5	0.166
EDS LV/antero-posterior dimension RV	1.9 ± 0.6	1.9 ±0.5	0.968
PVCs, thousands per day	15 [3.2; 35.5]	0.7 [0.01; 3.8]	<0.001
Non-sustained VT, %	55.3	26.3	0.021
Sustained VT, %	21.1	0	0.008
IST−			
LV EF, %	57.7 ±16.1	53.6 ± 10.6	0.018
EDS LV/antero-posterior dimension RV	2.0 ± 0.6	2.4 ± 0.5	0.091
PVCs, thousands per day	20 [0.6; 38]	3.7 [0; 9.3]	0.028
Non-sustained VT, %	71.4	16.7	0.046
Sustained VT, %	42.9	0	0.317

**Table 6 genes-16-00051-t006:** Baseline characteristics and endpoints of patients with DCM depending on the presence or absence of myocarditis.

Characteristic	Myocarditis+	Myocarditis−	*p*
N	34	26	-
Acute onset, n (%)	91.2	76.9	0.065
Relation to prior infection, n (%)	55.9	26.9	0.018
Myocardial morphological investigation, n (%)	47.1	15.4	>0.05
Pathogenic/likely pathogenic variants, n (%)	20.6	19.2	0.580
EDV, ml	182 [126; 233]	198 [142; 243]	0.584
LV EF (EchoCG), %	25 [20; 38]	33 [27; 41]	0.041
Interventricular septum, mm	9 [8; 10]	11 [9; 12]	0.024
NYHA CHF class ≥ 3, %	3 [2; 3]	3 [2; 3]	0.651
Administration of torasemide, %	50.0	23.1	0.031
PVCs, number per day	961 [160; 5227]	658 [6; 4281]	0.406
Non-sustained VT, %	58.8	34.6	0.054
Administration of amiodarone, %	67.6	46.2	0.059
ICD implantation, %	38.2	26.9	0.261
SCD, %	14.7	3.8	0.171
SCD + ICD interventions, %	29.4	15.4	0.168
Death, %	35.3	26.9	0.342
Heart transplantation, %	11.8	7.7	0.472

**Table 7 genes-16-00051-t007:** Baseline characteristics and endpoints of patients with RCM, depending on the presence or absence of myocarditis.

Characteristic	Myocarditis+	Myocarditis−	*p*
N (%)	5 (33.3)	10 (66.6)	-
Age, years	54.0 ± 17.5	49.4 ± 17.4	0.713
Acute onset, n (%)	0	1 (10)	0.667
Relation to prior infection, n (%)	2 (40)	2 (20)	0.758
Rhythm and conduction abnormalities			
PQ, ms	190 [180; 200]	160 [150; 160]	0.046
PVCs per day	5043 [827; 12 416]	326 [45; 1677]	0.086
Non-sustained VT, n (%)	3 (60)	4 (40)	0.427
Atrial fibrillation, n (%)	4 (80)	7 (70)	0.593
Parameters characterising heart failure			
NYHA CHF class	3 [3; 3]	3 [3; 3]	0.661
LV EF (EchoCG), %	47 [30.5; 59]	46.5 [43; 61.5]	0.759
LV EF (EchoCG) ≤ 45%, n (%)	2 (40)	4 (40)	0.713
E/A	3.6 ± 0.1	2.5 ± 0.3	0.076
Interventricular septum (EchoCG), mm	9.8 ± 1.9	10.6 ± 2.5	0.617
Posterior wall (EchoCG), mm	10.0 ± 2.2	10.2 ± 2.6	1.00
LV EDV (EchoCG), mL	84 [70; 125]	74 [64; 89.5]	0.327
RV (EchoCG), cm	2.5 [2.5; 4.2]	3.2 [2.8; 3.8]	0.639
Left atrium (EchoCG), mL	129 [63; 137.5]	106 [81; 153]	0.713
Right atrium (EchoCG), mL	120 [47.5; 136]	83 [50.5; 176.5]	0.739
End points			
Death, n (%)	0	2 (20)	0.429

## Data Availability

The data presented in this study are available upon request from the corresponding author due to privacy reasons.

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
