# Peer review of "Incidence and Impact of Myocarditis in Genetic Cardiomyopathies: Inflammation as a Potential Therapeutic Target"

_genes, 2025, doi:10.3390/genes16010051_

Round 1
Reviewer 1 Report
Comments and Suggestions for Authors
Authors wrote an interesting manuscript focused on myocarditis and cardiomyopathies. The study is well designed and data explained in a clear way. Some points to clarify:
1.- Please, all genes should be written in italic (line 46...)
2.- Please, the word "mutation" refers to any alteration in the genome. It should be more precise to use "rare variant" or "pathogenic variant"...
3.- ARVC is a disease included in a entity named ACM (Arrhythmogenic cardiomyopathy). Please update data focused in this global entity or add data concerning ALVC, biventricular cardiomyopathy, etc...
4.- Please clarify in the text if all cases showed a definite clinical diagnosis or suspected (borderline).
5.- Please add explanation about if more aggressive phenotypes showed myocarditis in comparison to mild or even soft phenotypes (in any of diseases).
6.- Mixed phenotypes showed more cases of myocarditis in comparison to sole phenotypes?
7.- Concerning genetic diagnosis, all rare variants identified after genetic analysis were classified as pathogenic/likely pathogenic following ACMG/AMP guidelines?
8.- Concerning myocarditis in tissue of any disease, which kind o cell were identified? The same types in all diseases? more neutrophils, B lymphocytes? T lymphocytes or may be macrophages?
9.- Apoptosis was also identified? In which diseases? These cases showed more/less inflammatory infiltrates in the tissue?
Reviewer 2 Report
Comments and Suggestions for Authors
The study is highly relevant and provides valuable insights into the relationship between myocarditis and genetic cardiomyopathies. However, significant clarifications, improvements in analysis, and presentation refinements are needed to ensure the study's robustness and clarity. Specifically:
The manuscript lacks information on adjustments for multiple comparisons, which is critical given the subgroup analyses, resulting in statistical rigor.
The exclusion of autoimmune and systemic inflammatory diseases needs justification, as this may introduce bias. It is unclear why certain systemic diseases such as sarcoidosis and autoimmune diseases were excluded. Given the inflammatory nature of myocarditis, excluding these populations might bias findings. Please justify these choices.
The claim that myocarditis primarily drives disease decompensation in ARVC lacks sufficient control data and comparative analyses.
Small subgroup sizes, particularly for restrictive cardiomyopathy, need to be acknowledged as a limitation.
Figure 5 needs clearer labeling and color consistency to improve readability.
The manuscript has the potential following significant revisions that address the statistical analysis, justification of methods, and improved interpretation of results. A stronger discussion of limitations and enhancements to visual clarity are also necessary to ensure the study's impact and scientific rigor.
Round 2
Reviewer 1 Report
Comments and Suggestions for Authors
No comments
Reviewer 2 Report
Comments and Suggestions for Authors
Authors have now improved the quality of the manuscript.